# Semantic-Layout-Guided Image Synthesis for High-Quality Synthetic-Aperature Radar Detection Sample Generation

**Yi Kuang** [1], **Fei Ma** [1]  , **Fangfang Li** [2,3], **Yingbing Liu** [1] **and Fan Zhang** [1,*]  

1  College of Information Science and Technology, Beijing University of Chemical Technology, Beijing 100013, China; mafei@mail.buct.edu.cn (F.M.); liuyb233@buct.edu.cn (Y.L.)
2  Aerospace Information Research Institute, Chinese Academy of Sciences, Beijing 100045, China; ffli1@mail.ie.ac.cn
3  Shandong Key Laboratory of Low-Altitude Airspace Surveillance Network Technology, Heze 274201, China
*  Correspondence: zhangf@mail.buct.edu.cn

**Abstract:** With the widespread application and functional complexity of deep neural networks (DNNs), the demand for training samples is increasing. This elevated requirement also extends to DNN-based SAR object detection. Most public SAR object detection datasets are oriented to marine targets such as ships, while data sets oriented to land targets are relatively rare, though they are an effective way to improve the land object detection capability of deep models through SAR sample generation. In this paper, a synthesis generation collaborative SAR sample augmentation framework is proposed to achieve flexible and diverse high-quality sample augmentation. First, a semantic-layout-guided image synthesis strategy is proposed to generate diverse detection samples. The issues of object location rationality and object layout diversity are also addressed. Meanwhile, a pix2pixGAN network guided by layout maps is utilized to achieve diverse background augmentation. Second, a progressive training strategy of diffusion models is proposed to achieve semantically controllable SAR sample generation to further improve the diversity of scene clutter. Finally, a sample cleaning method considering distribution migration and network filtering is employed to further improve the quality of detection samples. The experimental results show that this semantic synthesis generation method can outperform existing sample augmentation methods, leading to a comprehensive improvement in the accuracy metrics of classical detection networks.

**Keywords:** collaborative synthesis generation; image synthesis; diffusion model; SAR sample augmentation





## 1. Introduction

Synthetic-Aperture Radar (SAR) imaging technology, known for its high resolution, extensive coverage area, and independence from weather conditions, is extensively used in geographic information systems, resource exploration, and civil applications. With the development of deep learning, data-driven deep models are widely used in SAR processing tasks such as object detection and recognition [1–4].

In the realm of object detection tasks, data-driven deep learning techniques have showcased remarkable advancements in performance, surpassing traditional approaches such as template matching, image processing, and machine learning methods. Through extensive training on large-scale datasets, DNN-based methods have realized heightened precision and robustness in object detection capabilities. High-quality training samples constitute the foundational basis for achieving high-performance outcomes with deep learning approaches. However, due to challenges in acquiring SAR samples and annotating samples, SAR object detection encounters difficulties in matching the sample advantage seen in natural scene object detection. As a solution to this challenge, two technological approaches have emerged, namely few-shot learning and sample augmentation. In contrast to

few-shot learning, SAR sample augmentation represents a direct approach by augmenting and generating high-quality samples to fulfill the learning requirements of deep models.

Common techniques for SAR sample augmentation can be categorized into three groups: basic methods, image synthesis, and intelligent generation. The central premise focuses on augmenting object diversity while retaining the intrinsic characteristics of the samples. Basic methods include fundamental sample augmentation operations such as random transformations [5–9] and image erasure [10]. These basic methods enhance the diversity of SAR samples by simulating variations within them. While foundational methods enhance model robustness against diverse deformations in tasks like detection and recognition, their impact on the semantic distribution in SAR images is often marked by low background diversity and high semantic information may be limited. In such scenarios, these methods may hardly influence the object's semantic distribution and could inadvertently introduce noise into the augmented samples, leading to adverse impacts on model inference [11].

The image synthesis method enhances the semantic information of the sample by merging elements from various synthesized SAR images. Element synthesis techniques expand the SAR dataset by generating augmented samples through whole-image fusion [12–15] or element composition [16–20]. On the one hand, the semantic complexity of the samples has been increased by introducing auxiliary SAR images for fusion. For instance, Hiroshi Inoue et al. [21] randomly selected images from the dataset and fused them pairwise. The resulting synthetic samples significantly improved the performance of the classification network. Jiang et al. [12] adopted a different approach. They fused SAR scene images under various spatiotemporal conditions and trained a model using these synthetic samples. The model was able to effectively capture the changes in land cover in the images. On the other hand, by separating the objects and backgrounds within SAR images and placing sliced objects into backgrounds, new samples can be synthesized for semantic enhancement. This approach provides advantages in terms of both adaptability and cost efficiency. However, these synthesized images often overlook the diversity and complexity of real images, failing to account for factors like imaging direction and lighting angles. Moreover, abrupt changes in object edge intensities contribute to a lack of realism in synthesized images.

The intelligent generation method enhances SAR samples using deep generative models. This approach [22–26] exploits semantic constraints set by input conditions to guide sample generation, resulting in SAR images with crucial semantic information. Generative models are categorized into two types: likelihood-based generative models [27] and implicit generative models [28]. The former employ maximum likelihood criteria to learn the probability distribution of a sample, such as autoregressive models [29], flow models [30], and variational autoencoders [31]. In contrast, the latter do not explicitly represent probability distributions and include notable examples like generative adversarial networks (GANs) [32] and denoising diffusion implicit models (DDIMs). GANs generate realistic images via adversarial training between the generator and discriminator, while DDIM samples images iteratively from noise distributions. In recent years, generative models have exhibited remarkable performance in the field of natural image augmentation. However, generating high-quality SAR samples is challenging due to the limited quantity of SAR detection samples, their complex distribution characteristics, and the lack of annotated samples.

Despite the success of the above methods in achieving effective data augmentation for recognition tasks, they all have certain limitations that result in suboptimal performance in detection tasks. An inherent issue is that synthetic methods often overlook SAR images' distinct imaging mechanisms and traits, compromising the diversity and logical consistency of object positioning in synthesized images. Furthermore, generative models inherently possess randomness. In the context of object detection tasks, ensuring the accuracy of object categories in generated images to avoid label confusion adds complexity to sample augmentation. Additionally, generative methods often require rich conditional information

for guidance, and the creation of paired training samples demands significant human resources. In summary, SAR detection sample augmentation faces the following challenges:

- The synthesis of SAR images often overlooks the correlation between key elements, such as ensuring that objects do not overlap and maintaining logical layouts between objects and backgrounds. This lack of semantic information integration leads to a performance drop in real-world scenarios.
- The generation process of SAR detection samples lacks clear constraints on the range of semantic image transformations. As a result, the quality of generated images is unstable, making it difficult to ensure consistency in object categories. This renders the generated sample less suitable for high-reliability tasks.

To address the mentioned challenges, this study takes a semantic constraint approach from the SAR background and combines basic methods with intelligent generation techniques to optimize traditional image synthesis methods. Prior to sample synthesis, we performed semantic layout of backgrounds to ensure the diversity and logical coherence of the object distribution, and significantly improved synthesis efficiency. However, since image elements are derived from the original SAR sample library, there is no substantial variation in objects and backgrounds in the synthesized images. As a result, the diversity of augmented samples becomes limited in scope. Consequently, we further utilize generated synthetic samples to enhance the original dataset. Firstly, we leverage semantic layout maps to guide the training of a pix2pixGAN model, enabling semantic-controllable SAR scene augmentation. Moreover, we aim to feed the generative model with an increased volume of synthetic samples via progressive training to achieve higher-quality generated objects. At the same time, it is essential to clean the generated sample to acquire a high-quality augmented sample. We devised a sample cleaning strategy that combines distribution transfer and network-based filtering. This approach takes into consideration the statistical distribution characteristics of SAR samples and their corresponding labels, thereby effectively obtaining high-quality SAR samples.

In general, the main contributions of our research are as follows:

(1) A synthesis generation collaborative SAR sample augmentation framework is proposed. The diffusion model and SAR sample synthesis method are combined to achieve flexible and controllable SAR sample augmentation.

(2) A sample synthesis method based on a semantic layout is proposed. Prior to synthesizing objects into backgrounds, we pre-layout each background to avoid the inefficiency caused by repetitive evaluations of the same background during the synthesis process. Meanwhile, we employ a pix2pixGAN network trained on layout maps to achieve semantic-controllable background augmentation, further enhancing the diversity of the background in synthesized images.

(3) In order to obtain high-quality SAR-generated samples, a progressively trained conditional diffusion model is proposed, utilizing synthetic samples. By integrating the sample cleaning strategy of distribution transfer and network filtering, high-quality SAR samples are efficiently acquired.

The rest of this article is organized as follows. Section 2 provides a brief overview of related work on SAR sample augmentation. Section 3 presents a detailed description of our efficient sample augmentation strategy. In Section 4, we conduct extensive sample augmentation experiments and compare the results with other classical methods. Finally, Section 5 presents the conclusions of the experiments.

## 2. Related Works

### 2.1. Basic Methods of Image Augmentation

Fundamental image enhancement methods primarily include overall image transformations. These include information transformation and information fusion between images. These methods address image enhancement from various perspectives using direct and practical techniques. As summarized in Table 1, these methods offer diverse approaches to image enhancement.

**Table 1.** Basic methods of image enhancement and concise introduction.

| Methods | Transformations | Introduction |
|---|---|---|
| Information Transformation | Translation | Moving the image horizontally or vertically |
| | Rotation | Rotating the image at a certain angle |
| | Scaling | Stretching or compressing the image in a ratio |
| | Mirroring | Flipping the image horizontally or vertically |
| | Noise Addition | Adding noise to the image |
| | Cropping | Cropping different regions from the image |
| | Overlay | Overlaying multiple images together |
| | Illumination | Adjusting the brightness, contrast, and color of the image |
| Information Fusion | Image Stitching | Seamlessly merging multiple images together |
| | Image Blending | Blending two or more different images together |
| | Image Fusion | Fusing two or more different images together |

## 2.2. Sample Synthesis

Sample synthesis merges objects and backgrounds from varied sources, producing images with distinct distribution semantics [33]. The process of natural image synthesis can be broken down into two crucial steps: feature decomposition and feature recombination. Feature decomposition research is primarily anchored in image segmentation and matting [34,35], targeting enhanced segmentation accuracy. Conversely, feature re-combination research focuses on rectifying semantic inconsistencies during image synthesis, aiming for superior synthesized image quality. These semantic inconsistencies include appearance disparities (image incoherence) [36,37], geometric incongruities (unreasonable positions and shapes) [38,39], and logical incongruities (unreasonable element associations) [33].

Unlike natural imaging, SAR imaging remains impervious to lighting and atmospheric disturbances. Consequently, objects in SAR images typically exhibit a relatively consistent appearance, with minimal variations under different observational conditions. Furthermore, SAR objects typically manifest as reflection intensities. These differences are subtler compared to features like color and texture in visible images. Thus, during sample synthesis, an excessive focus on appearance consistency is not necessary. Given that SAR image backgrounds, often terrains or static objects, exhibit minimal temporal or weather-induced variations, the geometric consistency and logical interaction between backgrounds and objects are primarily manifested through their spatial positions. Therefore, SAR sample synthesis prioritizes bolstering the spatial relationship between objects and backgrounds for object detection, ensuring greater sample relevance and stability.

## 2.3. Sample Generation

The generation of detection samples refers to the production of diverse images under given conditional constraints [40]. The majority of SAR image generation is based on GAN networks. For sample augmentation, CGAN [41] integrates added conditional constraints into GANs, facilitating the generation of category-specific images. However, this approach suffers from issues of limited diversity in generated images and recurring problems of pattern collapse [42,43]. Following this, seminal models like Pix2Pix [44] and CycleGAN [45] were developed. Pix2Pix, as the earliest image-to-image transformation model, is founded on the principle of learning mapping relationships between paired input and output images. Conversely, CycleGAN tackles the issue of obtaining paired training samples. In recent years, diffusion models have gained great success in the field of natural image Artificial Intelligence-Generated Content (AIGC) due to their powerful creativity. Diffusion models consist of two processes: forward denoising and reverse inference. Jascha et al. introduced the reverse diffusion process in 2015, laying the groundwork for diffusion models [46]. In 2020, the debut of the denoising diffusion probability model, which integrates the denoising fraction matching function, brought diffusion models into prominence [47]. In the same year, Song et al. proposed the denoising diffusion implicit model [48], greatly improving the speed of reverse denoising. The distinct imaging techniques and inherent traits of

SAR samples have led to the infrequent application of diffusion models in SAR sample augmentation. However, Yuan et al. [49] have recently introduced an efficient, controllable training strategy tailored for conditional diffusion models.

## 3. Proposed Method

Figure 1 presents the schematic diagram of the proposed method. The augmented dataset comprises three subsets: $I_{S1}$, $I_{S2}$, and $I_G$. $I_{S1}$ denotes synthetic samples generated using a semantic layout strategy. $I_{S2}$ indicates synthetic samples augmented for background diversity, and $I_G$ corresponds to samples generated via the diffusion model.

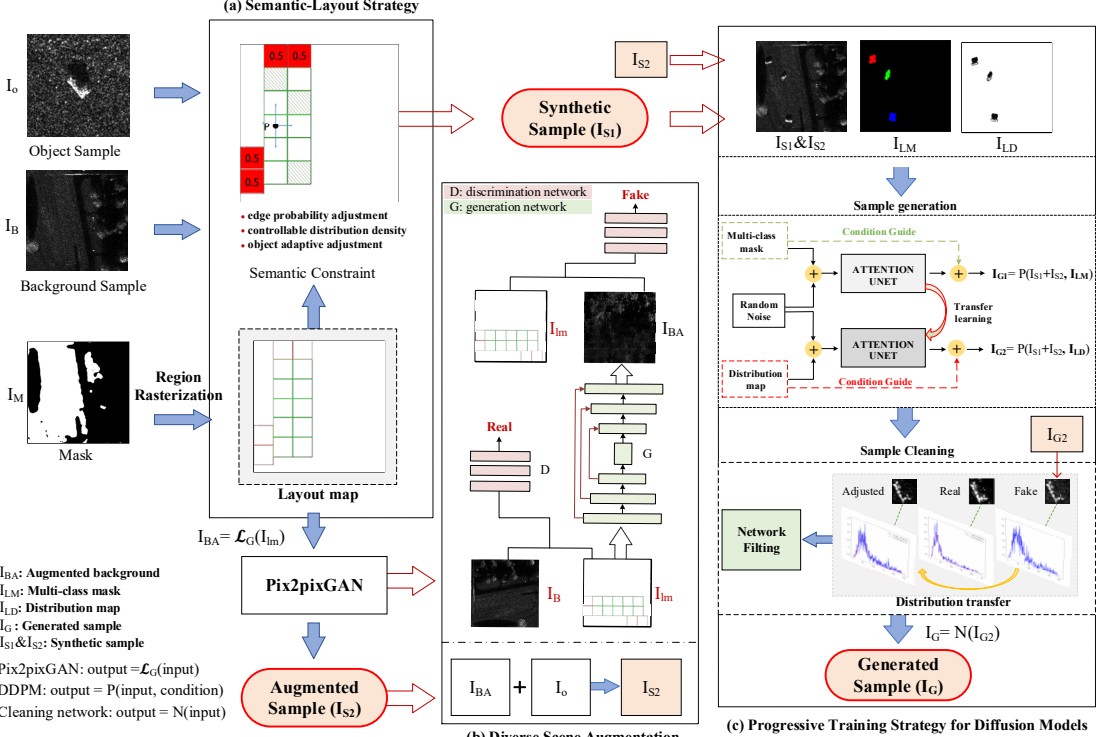

**Figure 1.** Proposed synthesis generation collaborative SAR sample augmentation framework. (**a**) The designed semantic layout strategy. (**b**) Diverse scene generation based on layout map. (**c**) Progressive training strategy for preserving more object details.

### 3.1. Semantic Layout Strategy

Our collection consists of n images, denoted as $I_{O1}, I_{O1}, ..., I_{On}$, sampled from the object library $O$. Additionally, we select a background image $I_B$, from the background library $B$. The process of image synthesis is represented as shown in Equation (1).

$$I_S = \sum_{i=1}^{n} \lambda_i I_{Oi} + I_B \tag{1}$$

where $I_S$ represents the synthesized image, and $\lambda_i$ is the positional parameter when synthesizing the objects $I_{Oi}$.

Due to the unique imaging mechanism of SAR images, choosing appropriate positional parameters $\lambda_i$ can significantly enhance the visual realism of synthesized images. The arbitrary selection of positions using land classification masks can result in over-regularization of $\lambda_i$, inefficiencies from redundant background determination, and potential time wastage due to inadequate positioning. To address this, we initially rasterize the mask to obtain refined positional parameters $\lambda'_i$. Then, guided by our semantic layout strategy, we allocate selection probabilities $p_i$ to each $\lambda'_i$, adjusting the intensity for object placements across various positions as $\sqrt{g_i}$. Here, $g_i$ is the ratio of the mean of pixel value within a rectangular

region (Mean(region)) to the mean pixel value of the entire background (Mean(global)). Consequently, our enhanced image synthesis approach is described as follows:

$$I_{S1} = \sum_{i=1}^{n} p_i \lambda'_i I_{Oi} * \sqrt{g_i} + I_B \tag{2}$$

$$g_i = \frac{Mean(region)}{Mean(global)} \tag{3}$$

### 3.1.1. Mask Rasterization

Layout grids are created via sliding window operations on predefined boxes. The effectiveness of these layout boxes is determined by assessing the area proportion of valid regions and their degree of dispersion, ultimately producing the scene's layout map.

In object detection tasks, sample augmentation often focuses on small objects or those with limited representation in specific classes. As a result, variations in object size are usually constrained. The dimensions of the layout boxes are set to match the largest object instance. The mask of the background image $B$, denoted as $M_b$, is also considered. Additionally, preset position boxes are defined as $a_i$. A column-major sliding window traversal is executed on the background image, as shown in Figure 2. $M_i$ is defined as the mapping region of $a_i$ on $M_b$, and the effective occupancy ratio $F_i$ of this region is calculated as shown in Equation (4):

$$F_i = area(\mathrm{m}_i) / area(\mathrm{M}_i) \tag{4}$$

where $F_i$ quantifies the proportion of the effective region $\mathrm{m}_i$ within the position box $M_i$.

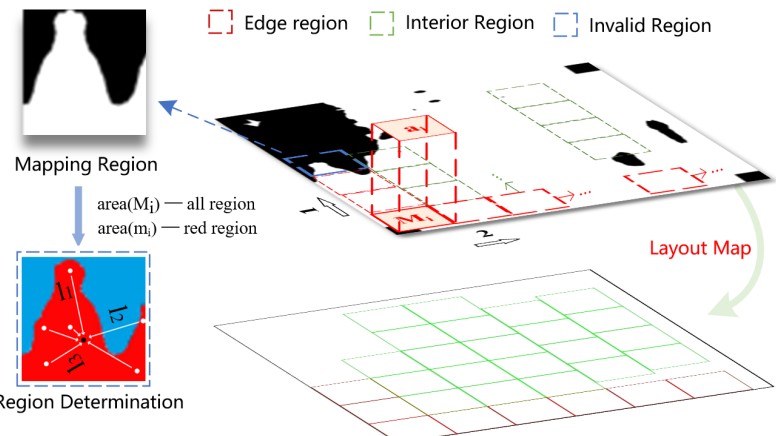

**Figure 2.** The process of mask rasterization.

When $F_i$ exceeds the threshold $\varepsilon$, indicating the object placement condition is met, we proceed to calculate the dispersion $D$ of the effective region within $M_i$. As described by Equation (5), we count the pixels in the effective region, denoted as n, and compute the centroid (X0, Y0) of this region. We calculate the mean squared Euclidean distances $l_i$ between these pixels and the centroid, determining the dispersion D of $M_i$. An appropriate dispersion threshold allows us to generate a refined set of layout position boxes.

$$D = \frac{\sum_{i=1}^{n} l_i^2}{n} \tag{5}$$

### 3.1.2. Semantic Constraints

Employed mask rasterization not only improves synthesis efficiency but also enhances the quality of the resultant images by imposing constraints on key semantic dimensions. The semantic layout strategy incorporates constraints in three key dimensions: object

positioning, distribution density, and scattering intensity. Specifically, it includes the following strategies.

To boost the robustness of the synthesized sample, edge mirroring augmentation is applied before initiating scene layout. Objects positioned at the edges are then cropped back, resulting in what we term 'incomplete objects'. To mitigate the adverse impact of too many 'incomplete objects', placement boxes are classified into two categories during scene layout: edge boxes (marked in red) and interior boxes (marked in green), as shown in Figure 2. The likelihood of successful object placement within edge regions is intentionally reduced. Furthermore, to prevent the clustering of small objects, the placement probability $p_i$ for each layout box is modulated. Specifically, the probability of placing an object successfully in edge regions is to be $0.5 \times p_i$.

$$p_i = e^{-\frac{d_i}{t}} \tag{6}$$

where, $d_i$ represents the distance from the center of each rectangle to the discrete center P, which is the center of the rectangle in the layout closest to the region center (X0, Y0). The hyperparameter $t$ is used to adjust $p_i$.

A greater distance between the ith rectangle and point P increases the likelihood of successful object placement, thus promoting object dispersion. Lastly, the scattering intensity for SAR objects is calibrated using the ratio $pl_i$. This ratio is derived from the mean pixel value $R_{mean}$ of the background for that region relative to the global background mean $B_{mean}$. The complete algorithm is shown in Algorithm 1.

---

**Algorithm 1:** The Semantic-Layout Strategy.

---

Input: background $B$ and mask $M$, object slice $I$
Output: layout map, position box information
Process:
1: The size of M is ($w_0$, $h_0$), perform edge mirroring augmentation to obtain $M_1$(w, h). Preset the size of position boxes as (m, n) and perform grid-based layout on $M_1$:
2: **for** j in 1, 2, ..., w **do**
3:        **for** k in 1, 2, ..., h **do**
4:                Calculate the effective occupancy ratio $F_i$.
5:                **if**    $F_i > \varepsilon$ **do**
6:                    Calculate the center point (X0, Y0) of the effective region.
7:                    Calculate the dispersion $D$ according to Equation (5).
8:                **end if**
9:                 Record the rectangular position information $\lambda'_i$, including center (x, y), width–height (h, w), and category (whether it is located on the edge)
10:        **end for**
11: **end for**
12: Count the number of position boxes (**num**), calculate the probability $p_i$ based on Equation (2).
13: start composing:
14:        **for** each B in the background dataset **do**
15:                Perform edge mirror augmentation on $B$ to match the same augmentation as $M$.
16:                Randomly select $I_{Oi}$ from the object library, where i $\in$ [1,**num**]
17:                Place the objects in the position boxes at random positions based on $p_i$.
18:                Synthesize samples according to Equation (**2**).
19:                Restore $B$ by cropping the edges.
20:        **end for**

---

### 3.2. Diverse Scene Generation

The Pix2pixGAN [44] model, built upon the foundation of Conditional Generative Adversarial Networks (CGANs) [41], is used for image-to-image translation. Its network structure is shown in Figure 3. Its core objective is to perform supervised translations between different image domains. Training Pix2pixGAN networks can be challenging, particularly with a scarcity of training samples. To overcome this, a series of strategies

are employed to simplify the complexity of cross-domain transformation tasks. One key strategy involves replacing the input mask with a layout map, streamlining the task's complexity. This approach helps alleviate the training difficulty, especially when dealing with limited samples. Additionally, the grid-based layout map provides extra spatial structure, aiding the network in learning the texture information within object regions.

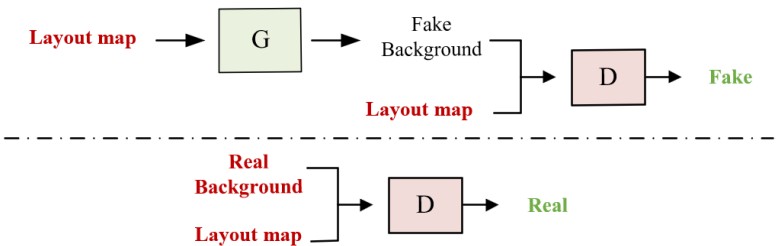

**Figure 3.** The architecture diagram of pix2pixGAN. G stands for the generator, and D is the discriminator.

To better control background generation, we introduced region-wise gradient loss ($L_{\text{laplacian}}$) and region position loss ($L_{\text{position}}$) into the generator's training, as shown in Equations (7) and (8).

$$L_{position} = \frac{1}{N}\sum_{i=1}^{N}|G(A)_i - B_i| \tag{7}$$

$$L_{laplacian} = \frac{1}{N}\sum_{i=1}^{N}\left(L(G(A)_i) - L(B_i)\right)^2 \tag{8}$$

where $N$ represents the total number of elements, $G(A)_i$ and $B_i$ denote the ith elements of the generated and the real images, and $L(.)$ is the Laplacian operator, used for computing Laplacian gradients of images.

Equation (8) calculates the Mean Square Error (MSE) between the Laplacian gradients of the generated and real images. Integrating these dual loss functions has proven effective in optimizing the generator's performance. These loss functions more accurately capture local structural nuances in the generated images, enhancing background generation quality. The refined background set is then used for sample synthesis, as shown in Equations (9) and (10).

$$I_{BA} = \mathcal{L}_G(I_{lm}) \tag{9}$$

$$I_{S2} = \sum_{i=1}^{n} p_i \lambda'_i I_{Oi} * \sqrt{g_i} + I_{BA} \tag{10}$$

where $\mathcal{L}_G(input)$ represents the pix2pixGAN model output, $I_{BA}$ is the augmented background, and $I_{S2}$ the synthesized sample with background augmentation.

*3.3. Progressive Training of Synthetic Data-Driven Diffusion Models*

3.3.1. Progressive Training of a Diffusion Model

In contrast to natural scene images, SAR scene clutter features a more intricate and diverse array of characteristics, encompassing various forms of interference, spatial correlations, and non-Gaussian statistical distributions. The Denoising Diffusion Probabilistic Model (DDPM) is a state-of-the-art generative model effective in capturing the complex clutter features of SAR scenes. It preserves spatial correlations and offers flexibility and controllability.

Noise addition in the diffusion model follows a Markovian procedure. In this model, $\alpha_t$ is the weight parameter for noise addition, and $Z$ represents Gaussian noise sampled from a standard normal distribution. In each successive step of the forward noise addition process, denoted as $X_t$, the computation involves noise addition weight $\alpha_t$ and Gaussian noise $Z_t$.

$$\begin{aligned} X_t &= \sqrt{\alpha_t}X_{t-1} + \sqrt{1-\alpha_t}Z \\ &= \sqrt{\delta_t}X_0 + \sqrt{1-\delta_t}Z_t \end{aligned} \tag{11}$$

where $\delta_t$ is the product of the noise addition weights from previous steps ($\alpha_t \cdot \alpha_{t-1} \cdots \alpha_2 \cdot \alpha_1$). In each forward diffusion step, standard Gaussian noise is added until the image is full noisy.

Conversely, the denoising process functions as a contrasting operation to the noise addition process, aiming to systematically eliminate noise. This process iteratively removes noise to generate a new image. The noise addition process derives $X_t$ from $X_{t-1}$, while the denoising process estimates $\widetilde{X}_{t-1}$ from $\widetilde{X}_t$. Moreover, the conditional diffusion model includes a conditional input c for constrained denoising. In our experiment, the multi-class mask and distribution map serve as the condition $c$, the model $f_\theta\left(\widetilde{X}_t, c, \delta_t\right)$ estimates the added noise $\delta_t$ in the forward diffusion. Therefore, the conditional diffusion model can be expressed as Equation (12).

$$\widetilde{X}_{t-1} = \frac{1}{\sqrt{\alpha_t}} \left( \widetilde{X}_t - \frac{1 - \alpha_t}{\sqrt{1 - \delta_t}} f_\theta\left(\widetilde{X}_t, c, \delta_t\right) \right) + \sqrt{1 - \alpha} \overline{Z}_t \tag{12}$$

Although diffusion models have shown excellent performance in image generation tasks, they require significant data support for training. Drawing inspiration from semi-supervised deep matting networks, we synthesis large datasets using masks instead of trimaps for model training, achieving high-quality matting masks [13]. Guided by this approach, we first use a synthesis method to obtain the constraint condition c, and then apply a diffusion model for image generation.

Our methodology begins with using synthesis techniques to create multi-class masks and distribution maps, as shown in Figure 4. In the multi-class mask, different pixel values represent various object categories. And the distribution map includes real object information, not just masked regions. We start with the multi-class mask for pre-training, focusing on constraining position and resolution information. Subsequently, the distribution map is used for transfer learning to further constrain the texture and category information. Through a progressive conditional learning approach, our model becomes capable of generating SAR samples with predetermined positions and categories.

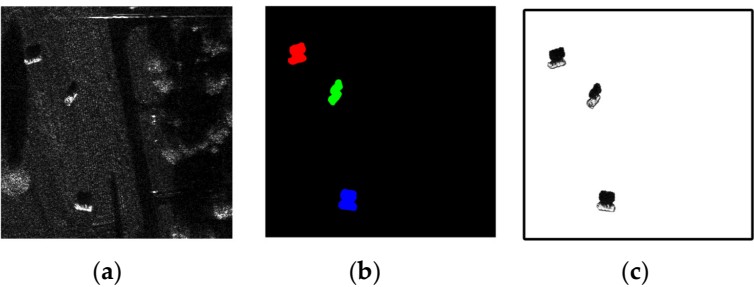

(a)          (b)          (c)

**Figure 4.** Input images for the progressive training strategy of the diffusion model. (**a**) Synthetic image, (**b**) multi-class mask, different colors indicate different categories, (**c**) distribution map.

### 3.3.2. Cleaning Generated SAR Samples

Outputs from the generative model can have issues like excessive background noise, object distortions, and semantic inconsistencies, reducing the utility of generated samples for augmentation tasks. Therefore, cleaning these generated SAR samples is essential to ensuring their quality and usability. SAR images often deviate from the conventional Gaussian distribution in pixel values with their intricate terrains and complex scattering mechanisms. Instead, their pixel values typically follow the K-distribution (Kappa distribution). The K-distribution effectively models these complex scattering mechanisms and noise characteristics, as illustrated in Equation (13).

$$f(x; k, \theta) = \frac{2k|x|^{\frac{2k-1}{2}}}{\Gamma(k)\theta^k} \exp\left(-\frac{|x|^k}{\theta}\right) \tag{13}$$

The K-distribution is defined by two parameters: $k$, representing degrees of freedom, and $\theta$, the scale parameter. Based on the labels of the synthetic sample, the slices of objects from the generated images are extracted. As shown in Figure 5, we fit these object slices to the K-distribution and adjust their pixel values based on discrepancies in the $k$ parameters. Next, we use a yolov5 model trained on the original samples to identify the generated samples. The detection results are cross-verified with actual labels to facilitate sample cleaning.

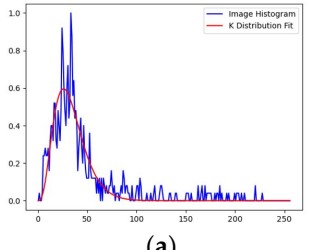 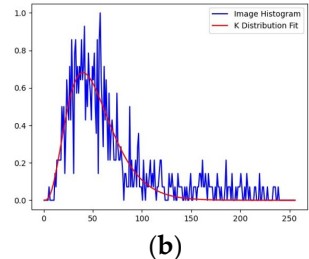 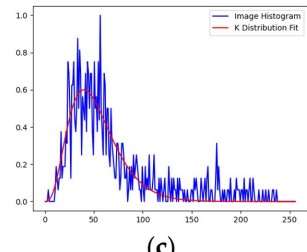

(**a**) (**b**) (**c**)

**Figure 5.** Transfer of distribution characteristics. (**a**) Original slice; (**b**) the generate slice; (**c**) distribution transfer.

## 4. Experimental Results and Analysis

This section introduces the SAR dataset used and compares our method with other SAR sample augmentation methods.

### 4.1. Experimental Data and Evaluation Indexes

(1) Experimental Data. To establish a robust baseline for sample augmentation, we used the MSTAR dataset to generate synthetic samples and created more representative and diverse samples for training and testing. We focused on ensuring the coherence of background and object resolution and the semantic authenticity of small object distribution. We selected 10 object classes, including 2S1, bmp2, BRDM_2, among others, and employed various image enhancement strategies. This process results in a dataset comprising 1000 high-resolution SAR images, serving as the baseline for our experiments. Moreover, we prepared a test set of 311 SAR images with corresponding masks, matching the specifications of the baseline dataset. This test set covered 10 different object types and 8 land cover types, such as barren land, grassland, farmland, shrubbery, forest, roads, buildings, and ocean.

(2) Evaluation Metrics. This experiment employed two primary sets of evaluation metrics to assess the performance of the generated images and the effectiveness of the augmented datasets for the given task. Initially, well-established quality evaluation metrics were used, such as the Inception Score (IS) and Fréchet Inception Distance (FID). They were utilized to assess image quality and quantify the quality of images generated by the models. Secondly, the precision (P), recall (R), and mean average precision (mAP) metrics were employed to evaluate the performance of the augmented datasets generated by different methods for a specific task. By comprehensively using these metrics, the experiment provides a more comprehensive assessment and comparison of various augmentation samples, ensuring more reasonable and accurate evaluation results.

- IS↑: Inception Score. IS consists of two components: the first value is the mean of the KL divergence, where a larger value indicates higher data quality. The second value is the standard deviation of the KL divergence, indicating richer sample diversity as the value increases.
- FID↓: Fréchet Inception Distance. FID is a metric used to measure the difference between generated images and real images. The lower the FID indicates the smaller difference between generated images and real images.

- P↑: Precision. The precision rate refers to the ratio of the truly positive samples among the positive samples detected by the model. The higher the precision rate indicates the lower the probability of the model predicting negative samples as positive samples.
- R↑: Recall. The recall rate refers to the proportion of truly positive samples detected by the model among all positive samples. A higher recall rate indicates a stronger ability of the model to correctly detect positive samples.
- mAP↑: mean Average Precision. The overall performance evaluation index of the model. A higher mAP value indicates superior model performance in object detection tasks.
- F1-score↑: F1-score is a performance metric that provides a comprehensive evaluation of the accuracy of a classification model by considering both precision and recall.

(3) Experimental environment. All experiments were executed on a uniform hardware configuration, with detailed environmental settings enumerated in Table 2.

**Table 2.** Basic experimental environment settings.

| Platform | Windows 11 |
| --- | --- |
| CPU | Intel Core i7-12700 k |
| Memory | 16 G |
| GPU | Nvidia GeForce RTX 3070 |
| Video memory | 8 G |

*4.2. Comparison with Other Advanced Methods*

To evaluate the quality of the augmented samples, we performed qualitative and quantitative comparisons with samples produced by leading generative networks. The models for comparison are as follows:

- Pix2pixGAN [44]: An improved generative adversarial network that achieves real-time transformation from input images to object images by learning the mapping relationship between them.
- CycleGAN [45]: An unsupervised generative adversarial network that maintains the original image's content using cycle consistency loss.
- BicycleGAN: By introducing additional consistency loss and mutual information loss, it enhances controllability and diversity in image generation tasks.
- DDPM [47]: A generative model that transforms random noise into a coherent image by simulating the reverse of diffusion, providing a unique way to create high-quality and diverse samples.

To evaluate the quality and label alignment of the generated sample, we conducted both qualitative and quantitative comparisons. The yolov5 model was employed to evaluate label alignment in the generated samples, with results detailed in Figure 6 and Table 3. The original training dataset consisted of 1000 images, and the test set contained 311 images. We generated 1000 images separately using three different GAN networks. The generated samples originated from three datasets: one synthesized with a semantic layout strategy (Ours-SL, 500 images), one blending synthetic backgrounds with real objects (Ours-BA, 500 images), and one created through progressive training of a diffusion model (Ours-GE, 1000 images). These datasets collectively form our augmented dataset (Ours-all).

Our qualitative and quantitative analyses revealed key insights: CycleGAN-generated images exhibit noticeable artifacts, particularly around object areas and suffer from limited diversity and low realism. In contrast, pix2pixGAN showed superior performance in image-to-image translation compared to CycleGAN. However, it tends to produce images with more background noise, lower resolution, and limited background diversity, evidenced by a KL (std) of only 0.038. While BicycleGAN excels in generating realistic backgrounds, it faces significant object distortions, resulting in a high FID score of 32.07. It also encounters issues of mode collapse in background generation. Compared to other GAN-based methods, DDPM generates higher-quality images, achieving a KL (mean) of 1.871 and a significantly lower FID score of 25.93. Nonetheless, it faces challenges in object generation distortions

and does not meet the label matching requirements for data augmentation. Our generative model (Ours-GE) outperformed the benchmarks, achieving an FID score of 25.54, and produced images with higher resolution and fewer object distortions. However, it does not match pix2pixGAN in terms of background complexity.

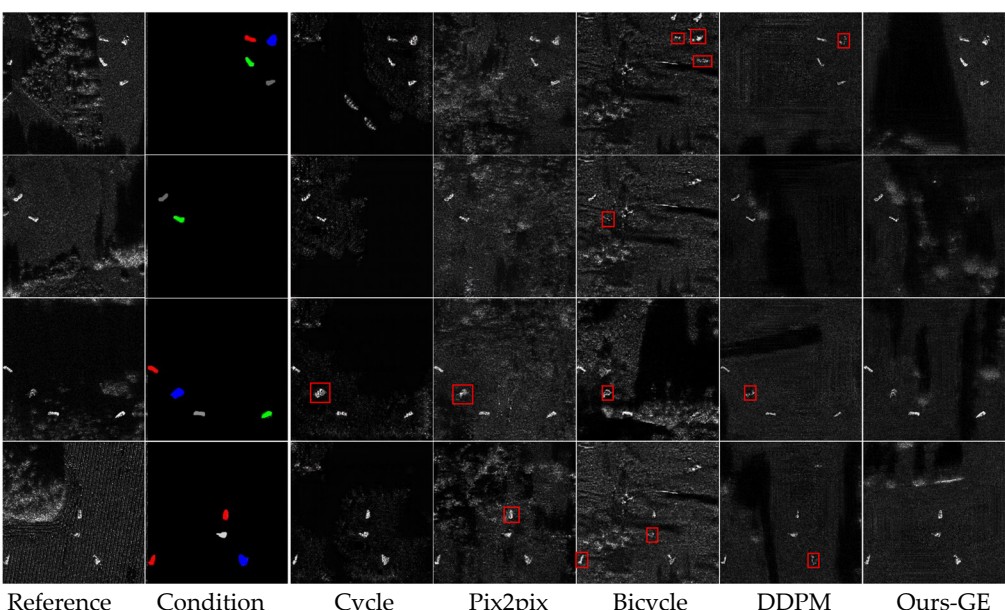

Reference    Condition    Cycle    Pix2pix    Bicycle    DDPM    Ours-GE

**Figure 6.** Qualitative comparison between Ours-GE and other generative models. There are four sets of example images, including five classes of MSTAR objects: 2S1, BMP2, BRDM_2, BTR_60, and BTR_70. The red boxes highlight severely distorted portions in the generated objects.

**Table 3.** Quantitative comparison of various augmented samples. Bold words represent the best results. ↑ and ↓ indicate the optimization direction of the indicator.

| Method | KL (Mean)↑ | FID↓ | KL (Std)↑ | Matching Degree (%)↑ |
|---|---|---|---|---|
| Original | 1.866 | - | 0.163 | 92.9 |
| CycleGAN | 1.349 | 28.42 | 0.070 | 18.5 |
| Pix2pixGAN | 1.252 | 28.47 | 0.038 | 15.1 |
| BicycleGAN | 1.126 | 32.07 | 0.056 | 10.2 |
| DDPM | 1.871 | 25.93 | 0.084 | 48.6 |
| Ours-SL | 1.782 | - | 0.010 | 92.7 |
| Ours-BA | 1.300 | 30.17 | 0.042 | 87.8 |
| Ours-GE | 1.774 | 25.54 | 0.090 | 78.2 |
| Ours-all | **1.884** | **18.83** | **0.180** | 81.8 |

To further validate the effectiveness of our augmented data, we trained a yolov5 network on the original dataset to assess the alignment between generated sample and object labels. In our study, we utilized the YOLOv5 model to perform detection on generated samples. Subsequently, we compared the detection results with ground truth labels, as depicted in Table 3. The term "Matching degree" is employed to indicate the degree of correspondence between the test results and the ground truth labels, represented as a percentage. As shown in the last column of Table 3, GAN networks perform poorly in generating specified-category small objects, with matching rates ranging from 10% to 20%. DDPM demonstrates improved performance in generating specified-class targets, though its matching percentage remains below 50%. In comparison, datasets synthesized using real objects and real backgrounds have nearly identical label matching rates to the original dataset. However, with the introduction of generated backgrounds in Ours-BA, there is a slight decrease in the alignment between synthesized samples and labels. Notably,

samples generated by Ours-GE achieve a label matching rate of 78.2%, indicating that they retain crucial feature information from real samples. This characteristic renders them highly suitable for sample augmentation purposes. Finally, our augmented dataset, which amalgamates three distinct enhancement techniques, exhibited improved sample diversity compared to the original dataset when evaluated using the KL divergence metric.

### 4.3. Ablation Analysis of Different Sample Augmentation Strategies

Throughout the experiments, we maintained consistent training parameters across all network models and utilized a uniform validation dataset. We evaluated the effectiveness of different augmentation schemes proposed in classic detection models Yolov5, RetinaNet, and VarifocalNet. This involved contrasting the detection results of different augmentation schemes on these classic networks.

(1) Semantic Layout Strategy. Our dataset, synthesized via a semantic layout strategy, demonstrated superior performance in multiple metrics when compared to a dataset generated through random placement across various parameters of RetinaNet and VarifocalNet, particularly achieving a notable 7.3% improvement in accuracy P for VarifocalNet. This is attributed to the fact that the semantic layout strategy rigorously constrains the distribution of objects on the background during instance placement, enhancing the association between objects and backgrounds. Consequently, the synthesized images demonstrate better semantic consistency between objects and backgrounds, enabling the model to distinguish between objects and backgrounds more effectively. However, for the yolov5 model, while we observed gains in Precision (P) and mean Average Precision (mAP), there was a decline in Recall. We speculate that this could be related to the dense prediction nature of the yolov5 network, where sparse object distribution may have adversely affected detection performance.

(2) Diverse Background Generation. Our comparative experiments (SL vs. SL + BA, as detailed in Table 4) indicated that background enhancement led to significant improvements across multiple detection metrics. As shown in Figure 7, the diverse background images generated from layout maps relax strict spatial constraints while ensuring accurate generation of land cover types within the position boxes. This flexibility allows for the generation of various terrains outside the position boxes. Consequently, the introduction of two additional loss terms, $L_{laplacian}$ and $L_{position}$, enhances the background details beyond the position boxes. This enhancement, specifically in texture and lighting, is depicted in Figure 7. After augmenting the background diversity, the detection network's feature map outputs during training are illustrated in Figure 8. By synthesizing diverse backgrounds with real objects, the model has been exposed to samples from various environmental contexts. This exposure has contributed to the enhancement of its robustness and its capacity to successfully differentiate objects from backgrounds in various environmental settings.

**Table 4.** Comparison of detection results on classical networks with different augmentation strategies.

| Network | Method | P (%) | R (%) | mAP (%) | F1 (%) |
|---|---|---|---|---|---|
| Yolov5 | Baseline | 78.8 | 79.8 | 66.6 | 79.30 |
| | Random | 80 | 80.3 | 67.3 | 80.15 |
| | Ours-SL | 81.1 | 80.1 | 68.7 | 80.60 |
| | Ours-SL+ BA | 82.9 | 81.5 | 69.7 | 82.19 |
| | Ours-all | **85.5** | **86.3** | **73.6** | 85.90 |
| RetinaNet | Baseline | 69.2 | 78.9 | 54.3 | 73.73 |
| | Random | 70.2 | 79.5 | 54.8 | 74.56 |
| | Ours-SL | 70.5 | 79.8 | 55.2 | 74.86 |
| | Ours-SL+ BA | 71.9 | 81.9 | 57.1 | 76.57 |
| | Ours-all | **73.2** | **83** | **58.3** | 77.79 |
| VarifocalNet | Baseline | 74.4 | 81.9 | 59.5 | 77.97 |
| | Random | 76.7 | 82.6 | 61.6 | 79.54 |
| | Ours-SL | 81.7 | 83.4 | 66.4 | 82.54 |
| | Ours-SL+ BA | 82.2 | 83.8 | 67 | 82.99 |
| | Ours-all | **85.6** | **84.6** | **70.4** | 85.10 |

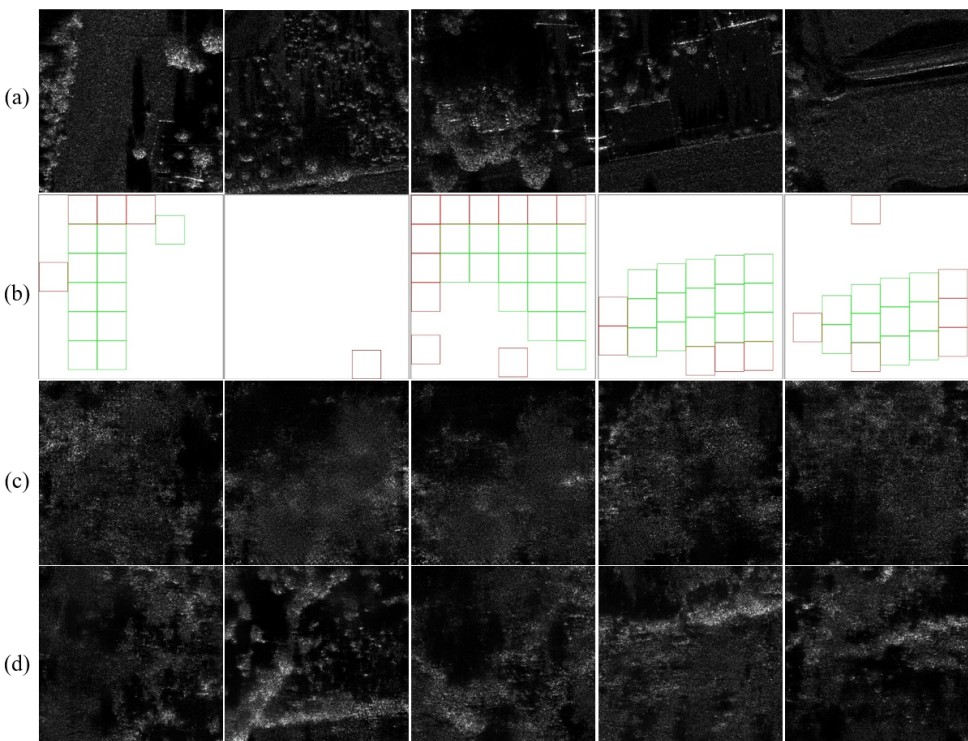

**Figure 7.** Examples of diverse background augmentation. (**a**) Displays five reference images selected from real backgrounds, encompassing various land cover types. (**b**) Represents our layout map. (**c**) Shows the results generated before improving the loss using pix2pixGAN, and (**d**) presents the generated results after improving the loss.

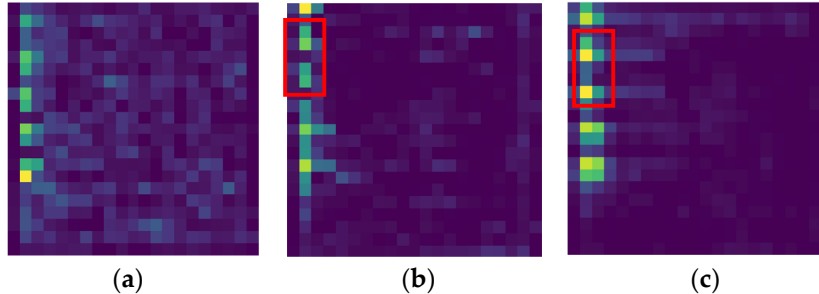

**Figure 8.** Comparison of the output feature maps before the SPP layer of the yolov5 network. The yellow pixels represent the most attended regions, followed by green and light green. The SL method reduces the network's focus on the background, the BA method further reduces the attention to the background and more accurately concentrates on the object regions. (**a**) Baseline dataset, (**b**) SL, (**c**) SL + BA.

(3) Progressive Training Strategy. Utilizing solely the multi-class mask as a constraining condition, the diffusion-model-generated images demonstrated commendable results, especially in object placement and overall scene composition in terms of object placement and scene generation. However, these images fell short in constraining object scattering intensity and preserving texture details, resulting in severe deformation and distortion issues in the generated samples, as shown in Figure 9a. Conversely, leveraging the distribution map as a guiding condition during transfer learning resulted in generated samples that more closely mirrored the appearance of genuine objects, as illustrated in Figure 9b, and possess power spectra closer to real SAR samples (Figure 10).

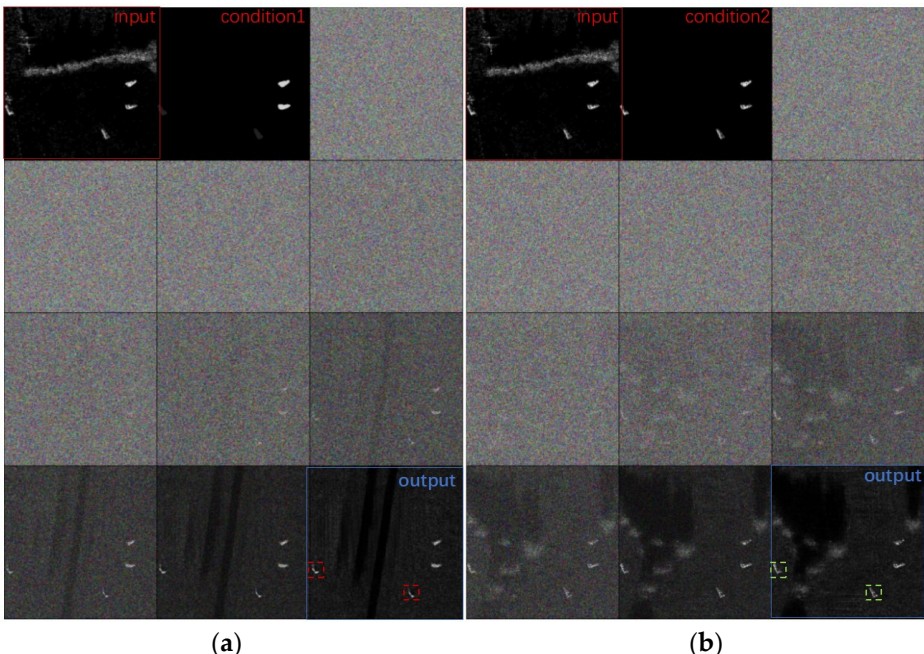

**Figure 9.** Visualization of the generated results. (**a**) Represents guidance using a multi-class mask, and (**b**) represents guidance using the distribution map. Compared to (**a**), the results generated using (**b**) contain more information about real objects and reduce deformation and distortion of objects.

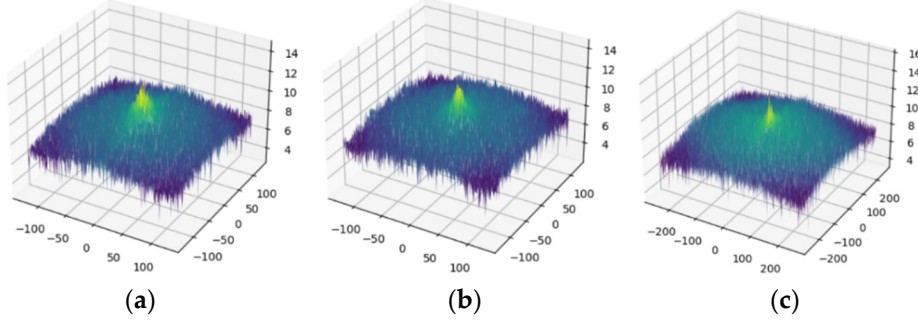

**Figure 10.** Power spectrum comparison. Compared to the power spectrum distribution of SAR sample images (**a**), the power spectrum of a sample generated by the single-step diffusion model (**b**) lacks side-lobe distribution. However, the power spectrum of our multi-condition progressive generated sample (**c**) is more concentrated in the low-frequency region, with more pronounced side-lobes.

We simultaneously observed a significant difference in frequency information between our generated samples and their real counterparts. Consequently, we applied K-distribution fitting to both real objects and generated objects separately. We then adjusted the pixel values of the generated objects in accordance with the K-distribution of real objects, aiming to retain more frequency information from the original data. Lastly, we utilized the yolov5 model for sample cleaning, inputting generated samples, comparing object detection results with real labels, and discarding samples showing significant discrepancies.

The study results indicate that generated samples, refined through progressive training and sample cleaning, surpass those from mere background augmentation in augmentation tasks. Our approach, Ours-SL + BA, resulted in an improvement of model accuracy by 0.5% to 1.8%, recall by 0.4% to 2.1%, mAP by 0.6% to 1.9%, and F1-score by 0.45% to 1.59%. Additionally, incorporating GE samples further improved model accuracy, with increases ranging from 1.3% to 3.4%, recall from 1.2% to 3.8%, mAP from 1.2% to 3.9%, and F1-score from 1.22% to 3.71%. These findings suggest that the generated images exhibit

significant semantic diversity, highlighting the overall improvement achieved through data augmentation in the context of the detection task.

*4.4. Discussion*

The proposed synthesis-generation collaborative method presents a novel framework for SAR sample augmentation. The method synthesizes samples effectively, ensuring visual consistency and logical coherence through semantic layout strategies. The integration of synthetic samples and generative models further enhances the semantic diversity of objects and backgrounds within the samples. However, the method faces challenges, particularly in improving its processing efficiency and versatility. On one hand, due to the method being primarily designed for MSTAR objects, it is less effective in generating large objects with complex structures and rich textures. On the other hand, the introduction of diffusion models brings relatively high computational costs. Therefore, developing lightweight approaches for this framework is crucial and will be our future research focus. Nonetheless, the sample augmentation method based on the mutual promotion of synthesis and generation still provides a new and meaningful research perspective for the remote sensing image processing field.

**5. Conclusions**

We propose a synthesis-generation collaborative SAR sample augmentation framework. Firstly, we introduce a semantic-layout strategy that enhances the correlation between objects and backgrounds through semantic constraints on layout boxes, resulting in semantically enhanced synthetic sample. To increase background diversity, we train a pix2pixGAN network to generate semantically controlled backgrounds. Finally, we progressively train a conditional diffusion model, coupled with distribution transfer and sample cleaning strategies, to obtain high-quality SAR samples. Comprehensive experiments indicate that our image synthesis intelligent generation framework outperforms existing sample augmentation approaches.

**Author Contributions:** Conceptualization, F.Z.; Methodology, Y.K.; Software, Y.K.; Validation, F.L. and Y.L.; Formal analysis, F.M.; Resources, F.M.; Data curation, F.L. and Y.L.; Writing—review & editing, F.M.; Supervision, F.Z.; Funding acquisition, F.Z. All authors have read and agreed to the published version of the manuscript.

**Funding:** This research was funded by National Natural Science Foundation of China, grant number 62271034.

**Data Availability Statement:** Data are contained within the article.

**Conflicts of Interest:** The authors declare no conflict of interest.

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
