# Peer review of "Semantic-Layout-Guided Image Synthesis for High-Quality Synthetic-Aperature Radar Detection Sample Generation"

_remotesensing, doi:10.3390/rs15245654_

Round 1

Reviewer 1 Report

Comments and Suggestions for Authors

The paper introduces a novel framework for SAR (Synthetic Aperture Radar) sample augmentation, which addresses the challenge of insufficient vehicle object samples in public SAR datasets. The framework combines semantic-layout guided image synthesis, pix2pixGAN network for background augmentation, and a progressive training strategy with Diffusion Models for scene clutter diversity. The authors claim improvements in mAP (mean Average Precision) of classical detection networks by 4%-11%.

Strengths:

1. The proposed synthesis-generation collaborative framework is a comprehensive approach to address the sample diversity and quality for SAR object detection.

2. The Semantic-Layout Guided Synthesis approach ensures the rationality of object locations and layout diversity, which is crucial for accurate object detection.

3. Utilization of Pix2pixGAN enhances background diversity and contributes to a more realistic sample generation.

4. Progressive training with diffusion models adds semantic controllability to the SAR sample generation, which can handle complex scene clutters.

Drawbacks:

1. The improvement is only measured regarding mAP, while other relevant metrics like precision, recall, or F1-score are not discussed.

2. The method might result in overfitting due to the heavy reliance on synthetic samples.

Recommendations:

1. Incorporate other relevant metrics like precision, recall, and F1-score to assess the framework's performance comprehensively.

2. Implement strategies to mitigate potential overfitting due to synthetic sample usage.

3. The practical implications of the framework should be discussed.

Author Response

All the comments have been addressed. Please check the attached file.

Reviewer 2 Report

Comments and Suggestions for Authors

This paper proposes a synthesis-generation collaborative SAR sample augmentation framework. Firstly, the authors propose a semantic-layout guided image synthesis strategy to generate diverse detection samples and utilize a pix2pixGAN network guided by layout maps to achieve diverse background augmentation. Then, a progressive training strategy of Diffusion Models is proposed to achieve semantically controllable SAR sample generation. Finally, a sample cleaning method considering distribution migration and network filtering is employed to further improve the quality of detection samples. The method appears reasonable, but some issues need to be clarified and modified. The main issues are as follows:

Major concerns:

1. Please explain what the 0.5 in the red color block and the green color block with a slash in the Semantic Constraint module in Fig. 1 represent.

2. In Formula 4, what dose L_{i} mean? Please explain.

3. In Section 3 and Section 4, some relevant references are not cited. For example, CGANS mentioned in Section 3 and CycleGAN mentioned in Section 4 do not mask their corresponding references.

4. In Formula 10, The "t" in \alpha _{t} should be at the subscript position of \alpha. Please correct the typos here.

5. In Section 4.2, the three methods compared in the experimental part are published in 2017. The proposed method should be compared with more state-of-the-art methods to demonstrate its effectiveness. For example, the methods are published in 2022 and 2023. Please supplement the experiments.

6. The authors should provide more explanations for the " Matching degree" mentioned in Table 3. And confirm whether you need to add "%" to it.

Comments on the Quality of English Language

Moderate editing of English language required.

Author Response

(The authors gave the same response as above.)

Reviewer 3 Report

Comments and Suggestions for Authors

To achieve flexible and diverse high-quality sample augmentation, the paper proposes a synthesis-generation collaborative SAR sample augmentation framework by combining the diffusion model and SAR sample synthesis method. The proposed approach sounds interesting, and the workload of this paper is full. However, there still have some issues:

1. Introduction part:

Line 58: This image synthesis methodshould be modified. The use of “this” is inappropriate.

Line 61: “On one hand” should be modified. The correct expression is “On the one hand…On the other hand”.

Line 137: What does “it” here refer to?

Line 139-143: “Section II, III, IV…” should be modified.

2. Related works part:

Line 185: It is recommended to add the full name of AIGC.

3. Proposed Method part:

Line 206-207: The grammar needs to be checked.

Line 220: The term “gi” is hard to understand. More explanations are necessary.

Line 271 Algorithm 1: The authors should explain how to obtain the rectangular position information λi. The authors also should check whether the Equation index (such as Line 12 in Algorithm 1) is correct.

Line 326: The meaning of the word “cc” is not clear.

4. The entire paper only uses the MSTAR dataset to display the results. It is recommended to add the modern SAR dataset to verify the migration and practicality of the model.

5 Lack of discussion on computational efficiency: The paper does not discuss the computational efficiency of the proposed method. It would be valuable to include information on the model’s inference time and resource requirements.

Comments on the Quality of English Language

It is recommended to further check and polish the language to improve the quality. 

Author Response

(The authors gave the same response as above.)

Reviewer 4 Report

Comments and Suggestions for Authors

 The paper addresses the challenge of data annotation in the context of ship detection in SAR domains. To address this challenge, the paper introduces a collaborative SAR sample augmentation framework, emphasizing synthetic data generation for enabling transfer learning using pix2pixGAN network, guided by layout maps. A progressive training strategy with Diffusion Models enhances scene clutter diversity, contributing to a more realistic representation. The proposed method demonstrates superiority over existing techniques, showing a 4% to 11% performance. The collaborative SAR sample augmentation framework proves helpful for detecting diverse land objects in SAR domains.  

  The paper reads well and the results are convincing. I have the following comments to be considered before publication:   1. I would like to see the adoption of this framework on a downstream task such as object detection directly to see how effective the method is.   2. Please release the codebase for this work to make results easily reproducible by other researchers. Comments on the Quality of English Language

The paper reads well.

Author Response

(The authors gave the same response as above.)

Round 2

Reviewer 2 Report

Comments and Suggestions for Authors

I recommend to accept this paper.

Comments on the Quality of English Language

None.

Reviewer 3 Report

Comments and Suggestions for Authors

In this paper, the authors focused the SAR sample augmentation problem, and proposed a valid and sound approach for this task. The paper has a good structure and the presentation is clear. I encourage the authors to make their implementations or demos publicly available, so that other research groups can reproduce the experiments and compare with each other.